



# Improving the Estimate of Higher Order Moments from Lidar Observations Near the Top of the Convective Boundary Layer

Tessa Rosenberger[1], David Turner[2], Thijs Heus[1], Girish Raghunathan[1], Timothy Wagner[3], and Julia Simonson[2,4]

[1]Cleveland State University, Cleveland, OH, USA
[2]NOAA Global Systems Laboratory, Boulder, CO, USA
[3]University of Wisconsin - Madison, Madison, WI, USA
[4]Cooperative Institute for Research in the Environmental Sciences, University of Colorado, Boulder, CO, USA

**Correspondence:** Tessa Rosenberger (t.e.rosenberger@vikes.csuohio.edu)

**Abstract.** Ground-based lidar data have proven extremely useful for profiling the convective boundary layer (CBL). Many groups have derived higher order moments (e.g., variance, skewness, fluxes) from high temporal resolution lidar data using an autocovariance approach. However, these analyses are highly uncertain near the CBL top when the depth of the CBL ($z_i$) is changing during the analysis period. This is because the autocovariance approach is usually applied to constant height levels and the character of the eddies are changing on either side of the changing CBL top. Here, a new approach is presented wherein the autocovariance analysis is performed on a normalized height grid, with a temporally smoothed $z_i$. Output from a large eddy simulation model demonstrates that deriving higher order moments from time series on a normalized height grid has better agreement with the slab averaged quantities than the moments derived from the original height grid.

## 1 Introduction

The atmospheric boundary layer (ABL) is the lowest portion of the atmosphere, typically ranging in depth from 10 m in extremely stable conditions to over 3 km, that interacts directly with the surface and is responsible for the majority of our weather. In particular, the ABL often has significant variability over the diurnal cycle due to the changing net radiation at the surface caused by the solar cycle. During the day when the surface is being heated by the sun, turbulent eddies rising from the surface create a well-mixed convective boundary layer (CBL) with turbulent eddies that range from approximately the size of the depth of the CBL (which will be denoted here as $z_i$) to sub-meter in size. Understanding and characterizing the properties of this turbulent CBL is critical to improve the modeling of transport and mixing within the CBL in weather and climate models.

Observations of turbulent mixing have been made for many dozens of years. Today's technologies include rapid response sonic anemometers and gas analyzers for in-situ observations, scintillometers for open path observations over larger volumes, and lidar observations from which profiles of turbulent moments can be derived. Higher order moments, such as the variance and skewness of a scalar as well as covariances between two geophysical variables (e.g., water vapor and vertical motion), are used to describe the turbulence in the CBL statistically. There are multiple areas where better understanding of these higher order moments is useful. For example, moisture variance in the CBL is important for understanding the boundary layer



moisture budget (Deardorff (1974); Lenschow and Wyngaard (1980); Huang et al. (2011)), the development of boundary layer clouds (Wilde et al. (1985); Golaz et al. (2002); Berg and Stull (2005)), and understanding the development of deep convection
(e.g., Berg et al. (2013)). Indeed, Wulfmeyer et al. (2016) Wulfmeyer et al. (2016) outlined a powerful approach that could be used to evaluate a wide range of similarity relationships that relate vertical gradients and mean profiles to turbulent moments using advanced ground-based lidar observations; similarity relationships often form the basis of turbulent parameterizations used within mesoscale and climate models.

Here, we focus on lidar observations of turbulent moments within the CBL. Lidar observations of water vapor (e.g., Muppa
et al. (2016); Turner et al. (2014)), temperature (Behrendt et al. (2015)), vertical motions (Berg et al. (2017); Lenschow et al. (2012)), aerosols (McNicholas and Turner (2014)), and fluxes (Behrendt et al. (2020); Kiemle et al. (2007); Senff et al. (1994)) have been used to derive higher-order moments in various locations. Lidar data, however, are frequently noisy due to both changing solar contributions and instrument noise, and thus separating out the atmospheric component to the higher order moments from the noise is challenging. Most lidar groups analyzing higher order moments use the autocovariance technique
pioneered by Lenschow et al. (2000) (2000; hereafter L-2000) to separate the two contributions, wherein the moments at lags above zero, which do not have any contribution from the instrument error which is assumed to be uncorrelated with time, are interpolated back to lag 0.

The L-2000 approach assumes that the turbulent nature of the CBL is not changing with time since statistics are derived from high-temporal resolution time series, given that the lidars are most often measuring very small volumes in the vertical column
above/below the lidar. Furthermore, as the larger eddies carry the most energy yet also occur less frequently, the time window analyzed must be sufficiently long to reduce the sampling error (Lenschow et al. (1994); hereafter L-1994). The two constraints provided by L-1994 and L-2000 restricts the analysis of lidar data to 1-to-2-hour periods when the CBL is quasi-stationary (i.e., where $z_i$ is not changing with time); these conditions are most commonly found in the mid-to-late afternoon (e.g., from 15:00 to 17:00 CDT in Figure 1).

However, there is a strong desire to be able to derive higher order moments from lidar observations when the CBL is rapidly evolving, such as the time period after the morning transition to when the CBL stops growing (e.g., from 10:00 to 15:00 CDT in Figure 1). Some studies have derived higher order moments from lidar data during periods when $z_i$ is rapidly changing; this is often done by using shorter time periods to derive the statistics (e.g., the 30-min window used in Berg et al. (2017)), which results in larger sampling uncertainties (per L-1994). Furthermore, all of these analyses are done on a fixed height grid; i.e., the
higher order moments are derived by looking at the time series at each range gate (height) observed by the lidar. This approach of using a fixed height grid from which to define the moments is insufficient at the top of the CBL when $z_i$ passes through the height layer being analyzed ($z_{anal}$) during the analysis period (i.e., when $z_i < z_{anal}$ early in the period and $z_i > z_{anal}$ at the end of the period).





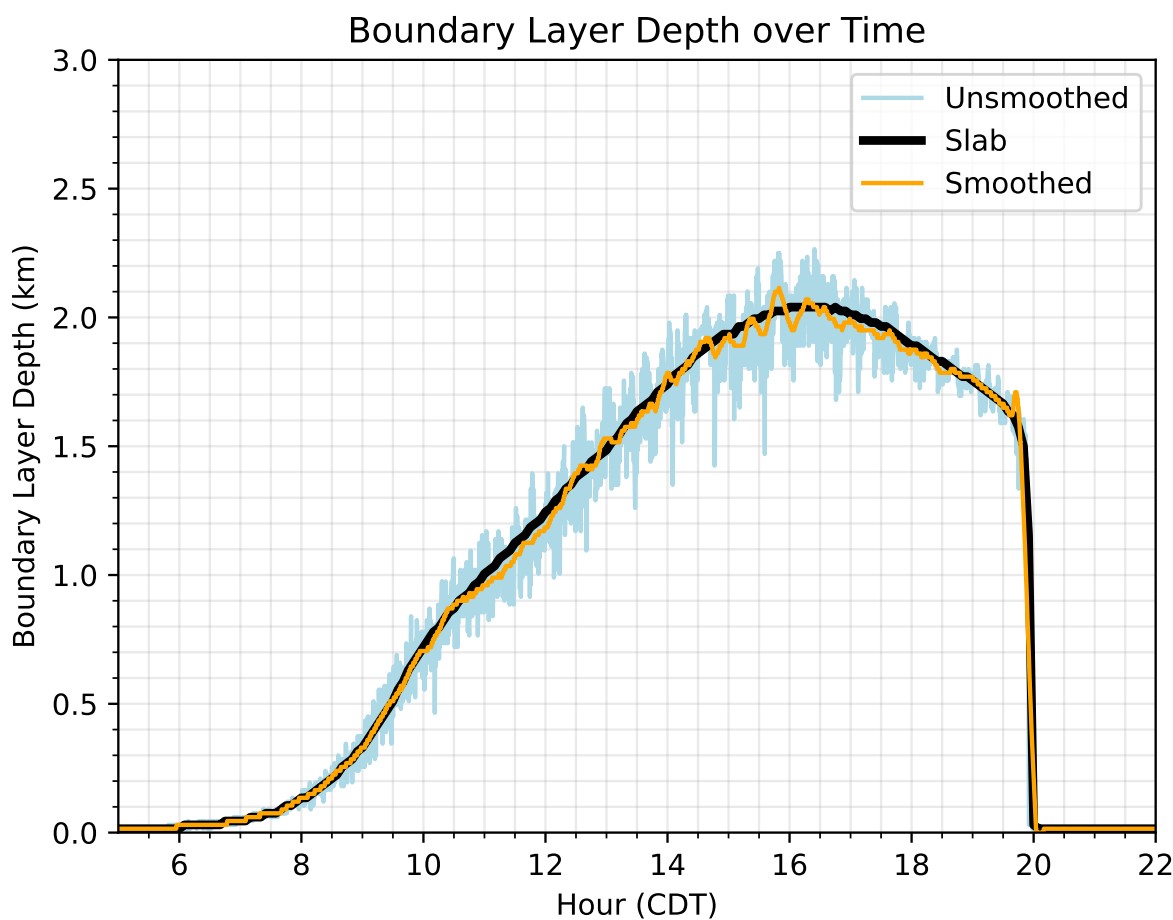

**Figure 1.** CBL depth over time, derived from the slab values (orange), from the instantaneous 10-s values of a single column (blue), and the 30- minute temporal average of the instantaneous values of the single column (green).

## 2 Approach

Our proposed approach is simple: instead of deriving higher order moments on a fixed height grid (z), the data are transformed to a normalized height grid ($\hat{z} = z/\overline{z_i}$) where the overbar indicated a temporal average. The advantage of this scheme is that, if $\hat{z} < 1 (> 1)$ for the entire analysis period then it is known that the time-series is entirely within (above) the CBL. This greatly simplifies the understanding of the statistics. The challenge thus becomes understanding the time period over which to derive $\overline{z_i}$ and demonstrating that computing the moments on the $\hat{z}$ grid is more accurate than using the regular z grid.

To investigate this, we utilized large eddy simulations of the CBL. The simulation used ARM's constrained variational analysis (VARANAL) for initial and boundary conditions (Xie (2017)). VARANAL yields values for surface fluxes, large-





scale advective and radiative tendencies that are spatially averaged over the entire ARM SGP domain. These simulations were performed with the MicroHH model (Heerwaarden et al. (2017)), using 25 m horizontal spacing over a 10 km by 10 km domain and 15 m vertical resolution. Statistics derived instantaneously over the entire domain were used as truth, and the lidar data were simulated by extracting out a high temporal resolution time-series as a single location in the middle of the domain. For this work, we will show results from 8 August 2017 over the Department of Energy Atmospheric Radiation Measurement (ARM) program (Turner and Ellingson (2016)) Southern Great Plains (SGP) site (Sisterson et al. (2016)). However, very similar results were found on other days, and these are not shown.

The evolution of the depth of the CBL (i.e., $z_i$) from the LES, derived three different ways, is shown in Figure 1. All three methods compute $z_i$ as the level of neutral buoyancy of a surface-based parcel. However, method 1 was the slab-averaged $z_i$ at each time t (i.e., over the entire model domain), method 2 was the instantaneous $z_i$ value for the selected column c (i.e., mimicking an instantaneous lidar observation) at time t, and method 3 used a Savitzky-Golay filter with a 1 h window to temporally average $z_i$ at that selected column c around time t. These will be denoted by $<z_i(t)>$, $z_{c,i}(t)$, and $\overline{z_{c,i}(t)}$ respectively.

For this work, we computed time-height cross-sections of variance and skewness of water vapor mixing ratio (q) from the LES output using three approaches: (a) using spatial statistics at each height level, which served as the truth dataset, (b) the baseline approach for a single column wherein the statistics were computed on a fixed z grid, and (c) the new approach for a single column where the statistics were computed on a normalized $\hat{z}$ grid using $\overline{z_{c,i}(t)}$, after which the moments were interpolated back to the regular z grid for comparison. We computed the variance and skewness at each level in the z or $\hat{z}$ grid by first extracting out the time series at that level for the time period being analyzed and detrending it. The variance is then computed as

$$Var(q) = \frac{1}{N}\Sigma(q - \overline{q})^2 \qquad (1)$$

the skewness as

$$Skew(q) = \frac{\Sigma(q - \overline{q})^3}{(N-1)(Var(q))^{\frac{3}{2}}} \qquad (2)$$

and the kurtosis as

$$Kurt(q) = \frac{\Sigma(q - \overline{q})^4}{(N-1)(Var(q))^2} \qquad (3)$$

where N is the number of points in the analysis window.

Note that we did not use the L-2000 technique here, as we did not attempt to simulate a true lidar observation by superimposing any random error. The primary purpose of this study is to demonstrate that using the normalized $\hat{z}$ grid provides more accurate measures of the variance and skewness than using the regular z grid, even though the former includes a contribution from the interpolation error that was introduced by putting the data on the $\hat{z}$ grid.



## 2.1 Results

A comparison of the q variance from the three calculation methods is shown in Figure 2. The slab value results (left) are truth to which the other two methods are compared. The slab values show that the variance is the highest at the top of the boundary layer from 1000 – 1730 CDT, after which it tapers off. Below the boundary layer top, the variance is much smaller. Both methods capture the higher variance along the top of the boundary layer, but the normalized $\hat{z}$ grid has less of a gap just before 1500 CDT while the regular grid has a more significant gap there. This tells us that the normalized $\hat{z}$ grid captures the variance better than the regular grid. Figure 3 shows the difference between the slab values and the regular grid (left) and the normalized $\hat{z}$ grid (right). These show that both methods are close to the slab values except for at 1230 CDT and 1500 UTC along the top of the boundary layer.

The q skewness is compared in Figure 4. These figures clearly show that the normalized $\hat{z}$ grid values are closer to the slab values in both magnitude and shape. Again, turning our attention to the values at the top of the boundary layer from 1000 – 1730 CDT, there is high skewness in a very thin layer. The regular grid underestimates the magnitude of the skewness here and overestimates the size of the layer with those highest skewness values. At the surface, both methods show higher levels of skewness than the slab at 1500 CDT and beyond. Figure 5 reiterates these points, showing that there is a greater difference between the regular grid results than the normalized grid at the top of the boundary layer, while at the surface, the differences are similar.

In the case of water vapor flux, the different grid methods must be applied to both q and w. The results for the flux $(\overline{q'w'})$ are shown in Figure 6. There are some clear differences between both grid methods and the slab values. The maximum flux is significantly higher than the slab values, and the two methods do not capture the flux well before 1230 CDT, especially in the middle of the boundary layer. The difference plots (Figure 7) show that both methods match with the slab values in the early morning and in the late afternoon, but not as much from 1000-1730 CDT, except for right along the top of the boundary layer, where it is very close to the slab values.

To better quantify the differences between these two methods and the slab values, Figures 8-10 show line plots of the variance (Figure 8), skewness (Figure 9), and water vapor flux (Figure 10) at ninety percent of the boundary layer (a), and top of the boundary layer (b), along with their respective root mean square errors (RMSE) calculated over the time window of 800 – 1800 CDT. We see that for the variance (Figure 8), and at both depths, the normalized $\hat{z}$ grid RMSE value is lower than the regular grid RMSE value, which shows that the normalized $\hat{z}$ grid method better captures the variance towards the top of the boundary layer than the regular grid method does. For the skewness (Figure 9), at 90% of the boundary layer (Figure 9a) and the top of the boundary layer (Figure 9b), the normalized grid method is significantly better than the regular grid method. Finally, looking at the flux (Figure 10), the methods are almost the same at 90% of the boundary layer (Figure 10a), and the normalized $\hat{z}$ grid method is better at the top of the boundary layer (Figure 10b).

We extended these methods to the case of the fourth moment, kurtosis, and calculated the variance, skewness, and kurtosis of vertical velocity (w). Tables 1-3 compare the RMSE values of the regular and normalized $\hat{z}$ grid methods for calculating variance, skewness, and kurtosis of q and w as well as the temperature $(\overline{\theta'w'})$ and moisture fluxes $(\overline{q'w'})$ at various heights





throughout the boundary layer ($0.75z_i$, $0.9z_i$, and $z_i$) In these tables, the values that are lower by the standard error of the two or more are bolded to show the better value. We found that at $0.5z_i$, neither grid method stood out as better as the RMSE values were either effectively the same or the values were better for an equal number of calculations, so we turn our attention to heights closer to the boundary layer depth. At $0.75z_i$ (Table 1), the normalized $\hat{z}$ grid method is better for every calculation except q variance and q kurtosis, where the regular grid method is better. For $0.9z_i$ (Table 2), the normalized $\hat{z}$ grid method is better in all cases except w skewness, where the methods are the same. Finally, at the top of the boundary layer (Table 3), the normalized $\hat{z}$ grid method is better in all cases except q variance, where the methods are the same. At every height, the normalized $\hat{z}$ grid method was better for q skewness. At depths closer to the boundary layer depth, the importance of the normalized $\hat{z}$ grid method for more accurate calculations becomes increasingly clear.

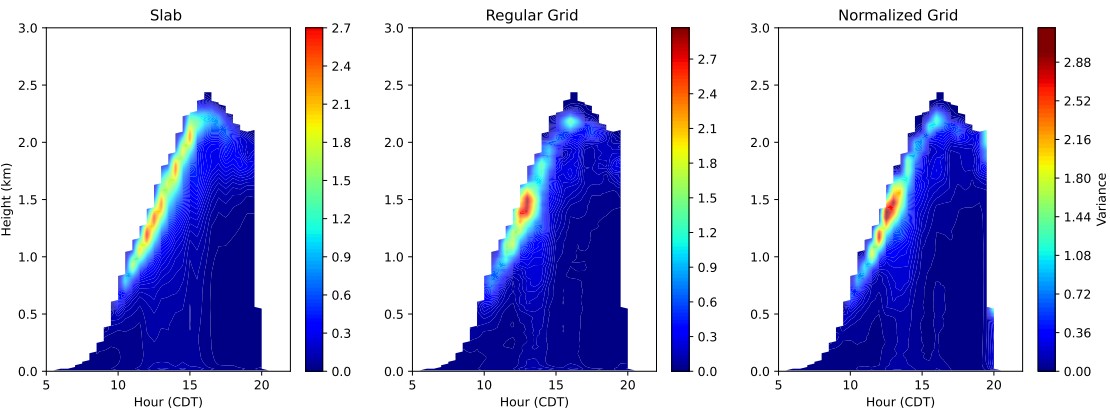

**Figure 2.** Time-height cross sections of q variance [units of $(gkg^{-1})^2$], computed from slab values at each height (a), on a regular z grid (b) and on the normalized $\hat{z}$ grid (c). Both (b) and (c) are averaged over a 1-h period centered on each 30-min.

**Table 1.** Comparison of the Root Mean Squared Errors for the regular z grid and normalized $\hat{z}$ grid methods, calculated over the time window of $800-1800$ CDT, at $0.75z_i$. Bolded values indicate markedly smaller RMSE value than the other height grid.

| Moment | RMSE Regular z Grid | RMSE Normalized $\hat{z}$ Grid |
|---|---|---|
| Variance $(q')[(gkg^{-1})^2]$ | **0.097** | 0.103 |
| Skewness $(q')[unitless]$ | 0.560 | **0.451** |
| Kurtosis $(q')[unitless]$ | **1.189** | 1.832 |
| Variance $(w')[(ms^{-1})^2]$ | 0.261 | 0.254 |
| Skewness $(w')[unitless]$ | 0.517 | 0.508 |
| Kurtosis $(w')[unitless]$ | 2.014 | **1.923** |
| Water Vapor Flux $(\overline{q'w'})[(gkg^{-1})(ms^{-1})^{-1}]$ | 0.078 | **0.076** |
| Temperature Flux $(\overline{\theta'w'})[(Km)s^{-1}]$ | 0.021 | **0.018** |



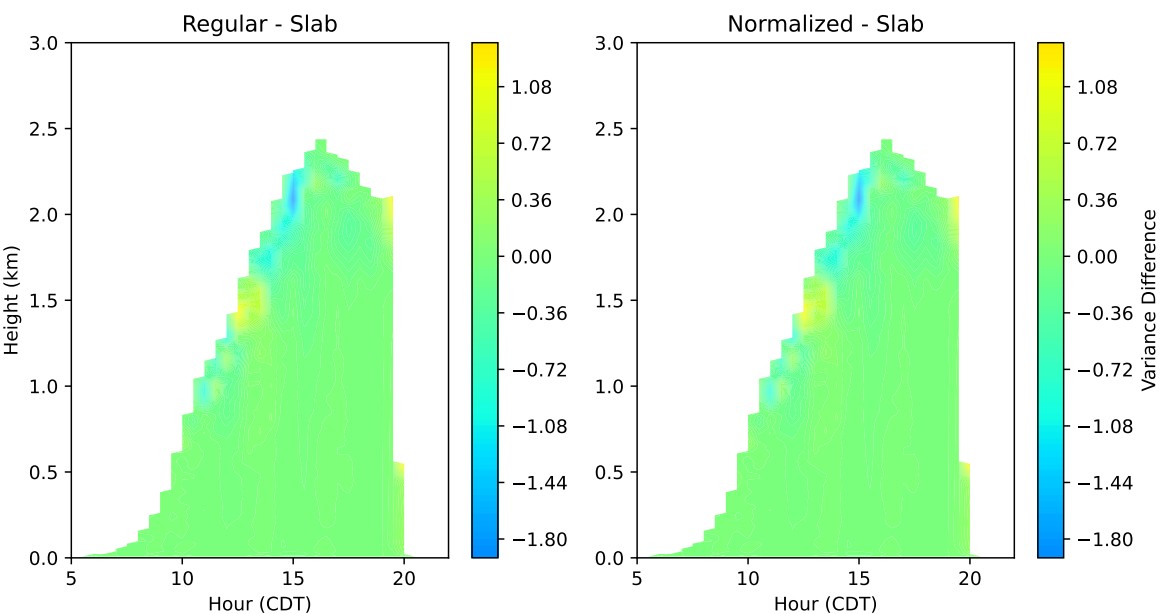

**Figure 3.** Time-height cross-section of the difference of the q variance [units of $(gkg^{-1})^2$]: Left - (panel 2b) minus (panel 2a) and right - (panel 2c) minus (panel 2a)

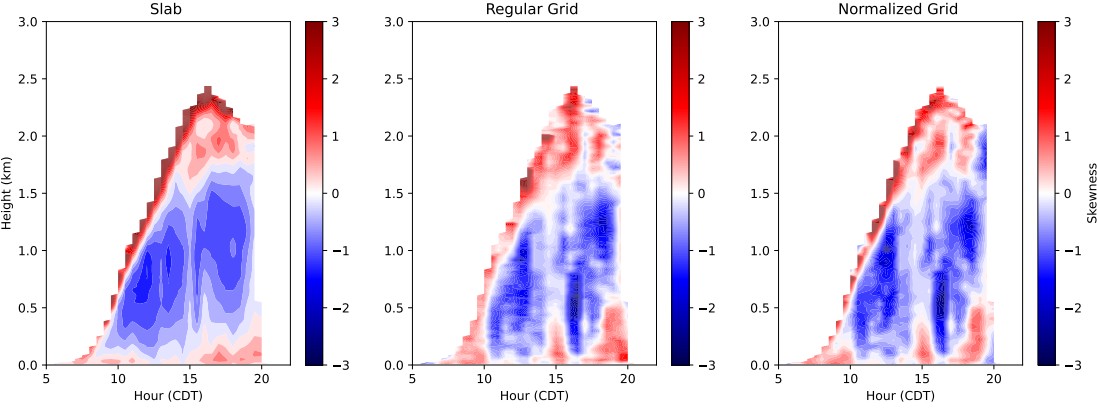

**Figure 4.** Time-height cross sections of q skewness [unitless], computed from slab values at each height (a), on a regular z grid (b) and on the normalized $\hat{z}$ grid (c). Both (b) and (c) are averaged over a 1-h period centered on each 30-min.



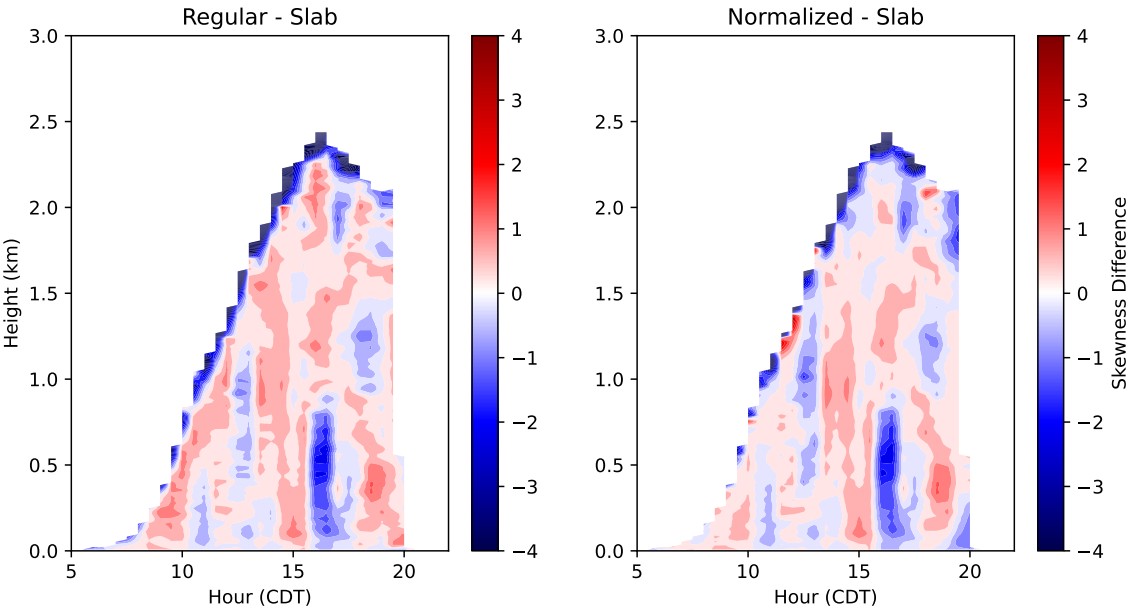

**Figure 5.** Time-height cross-section of the difference of the q skewness [unitless]: Left - (panel 4b) minus (panel 4a), and right - (panel 4c) minus (panel 4a)

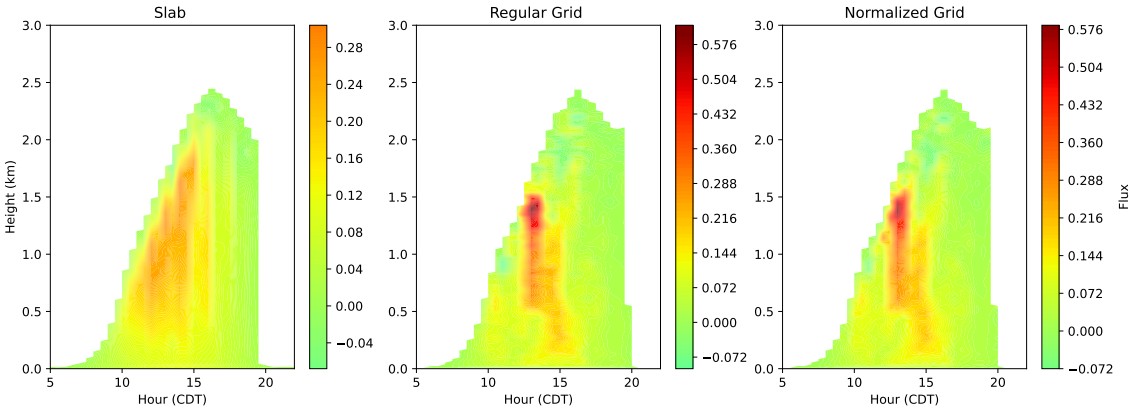

**Figure 6.** Time-height cross sections of q flux $(\overline{q'w'})$ [units of $(gkg^{-1})(ms^{-1})^{-1}$], computed from slab values at each height every 30 minutes (a), on a regular z grid (b) and on the normalized $\hat{z}$ grid (c). Both (b) and (c) are averaged over a 1-h period centered on each 30-min.

### 2.1.1 Discussion

The normalized $\hat{z}$ grid more accurately captures the q and w variance, skewness, kurtosis and temperature and moisture fluxes, especially at heights approaching the top of the boundary layer. By accounting for changes in the boundary layer over time,





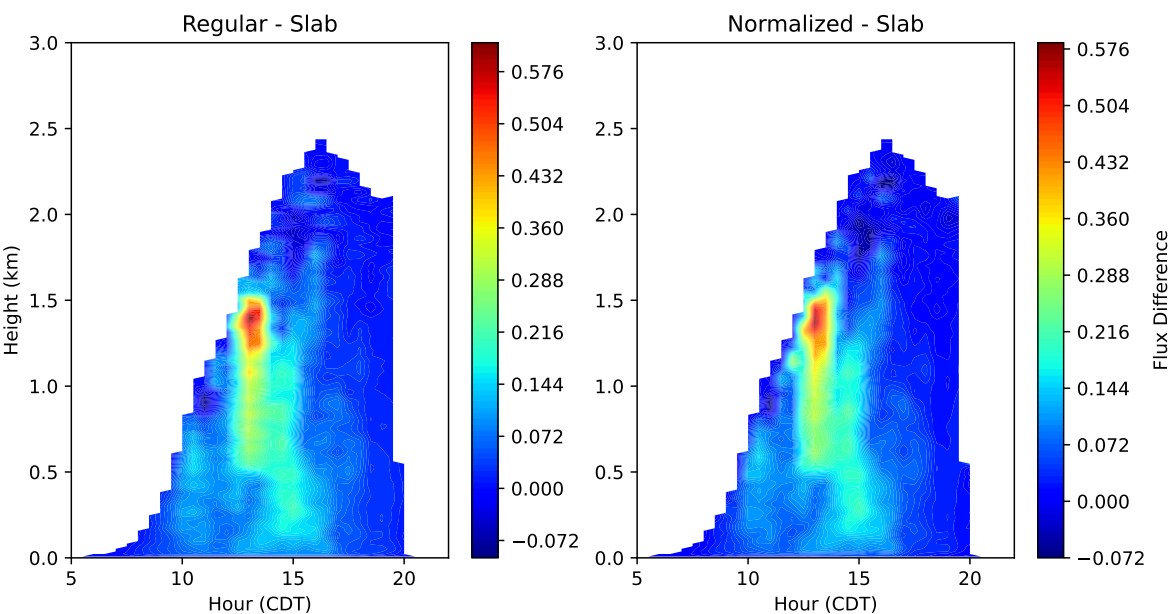

**Figure 7.** Time-height cross-section of the difference of the q flux $(\overline{q'w'})$ [units of $(gkg^{-1})(ms^{-1})^{-1}$]: Left - (panel 6b) minus (panel 6a) and Right - (panel 6c) minus (panel 6a)

**Table 2.** Same as Table 1 but at $0.9z_i$

| Moment | RMSE Regular z Grid | RMSE Normalized $\hat{z}$ Grid |
|---|---|---|
| Variance $(q')[(gkg^{-1})^2]$ | 0.301 | **0.259** |
| Skewness $(q')[unitless]$ | 0.547 | **0.379** |
| Kurtosis $(q')[unitless]$ | 1.222 | **0.752** |
| Variance $(w')[(ms^{-1})^2]$ | 0.179 | 0.179 |
| Skewness $(w')[unitless]$ | 0.639 | **0.598** |
| Kurtosis $(w')[unitless]$ | 2.537 | **2.436** |
| Water Vapor Flux $(\overline{q'w'})[(gkg^{-1})(ms^{-1})^{-1}]$ | 0.119 | **0.112** |
| Temperature Flux $(\overline{\theta'w'})[(Km)s^{-1}]$ | 0.046 | **0.044** |

this approach allows for a more accurate analysis of turbulence characteristics, particularly while the CBL is actively growing. This was particularly true for skewness, suggesting that higher order moments would benefit more from this new approach. These results are consistent across multiple analysis days (not shown).

Previous work that only considers a regular grid could be reanalyzed to be more accurate with this method. In the future, this method can be used for more accurate lidar analysis of the CBL turbulent statistics during the late morning transition. A





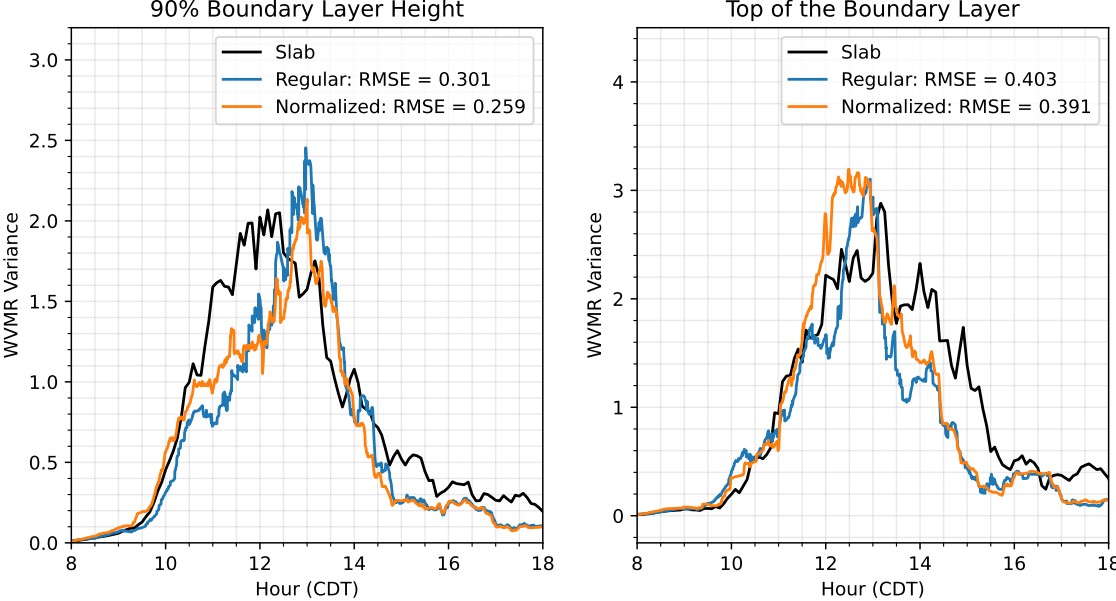

**Figure 8.** Line plot comparison of q variance [units of $(gkg^{-1})^2$] at 90% of the boundary layer (a), and at the top of the boundary layer (b) for the three computation methods: slab (black), regular grid (blue), and normalized $\hat{z}$ grid (orange), along with their respective Root Mean Square Error (calculated over the time period of 800-1800 CDT) with respect to the slab value results.

**Table 3.** Same as Table 1 but at $z_i$

| Moment | RMSE Regular z Grid | RMSE Normalized $\hat{z}$ Grid |
|---|---|---|
| Variance $(q')[(gkg^{-1})^2]$ | 0.403 | 0.403 |
| Skewness $(q')[unitless]$ | 1.124 | **0.623** |
| Kurtosis $(q')[unitless]$ | 4.124 | **3.094** |
| Variance $(w')[(ms^{-1})^2]$ | 0.205 | **0.202** |
| Skewness $(w')[unitless]$ | 0.525 | **0.473** |
| Kurtosis $(w')[unitless]$ | 2.644 | **2.466** |
| Water Vapor Flux $(\overline{q'w'})[(gkg^{-1})(ms^{-1})^{-1}]$ | 0.095 | **0.087** |
| Temperature Flux $(\overline{\theta'w'})[(Km)s^{-1}]$ | 0.047 | **0.046** |

larger time window could be used, since the changes in the boundary layer are already considered in the analysis, which will reduce the sampling error relative to previous studies done on a regular grid.

Further refinement is still necessary to determine optimal analysis periods guided by L-1994. Additionally, this method also would need to be adjusted for extremely rapid changes in $z_i$, such as during the evening transition. In cases where neither grid



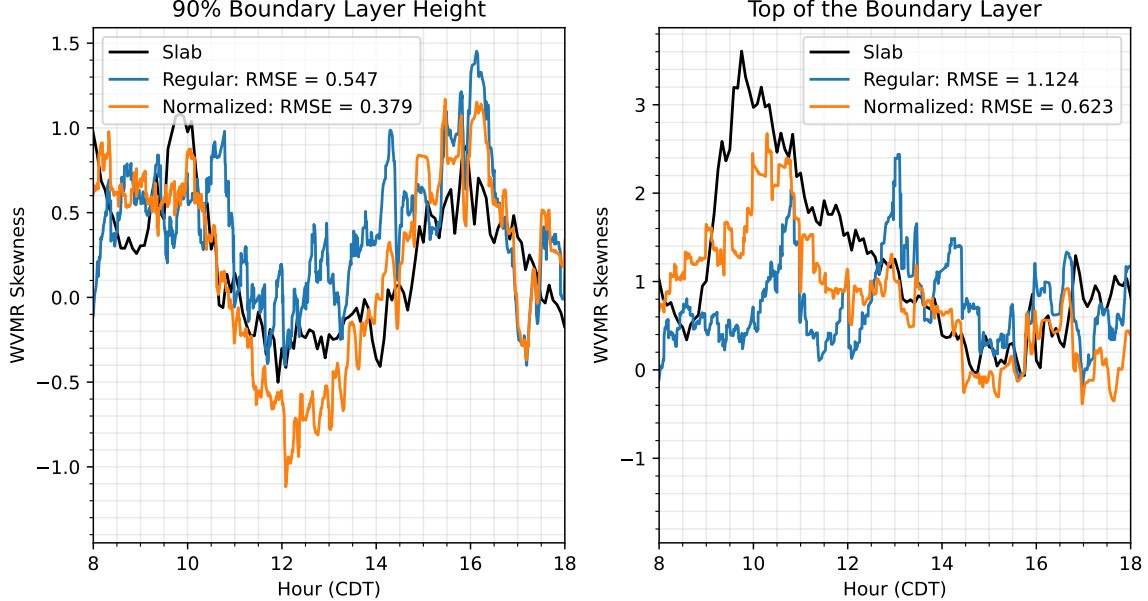

**Figure 9.** Line plot comparison of q skewness [unitless] at 90% of the boundary layer (a) and at the top of the boundary layer (b) for the three computation methods: slab (black), regular grid (blue), and normalized $\hat{z}$ grid (orange), along with their respective Root Mean Square Error with respect to the slab value results.

accurately captures the slab values, we must remember the spatial variability that a single column will never be able to capture (i.e., sampling errors). It is clear, especially in the variance and flux around 1230 CDT, that the single column is experiencing an updraft that is not representative of the entire domain. Further work must be done to reduce the impact of spatial variability.

## 3 Conclusions

This work shows that using a normalized $\hat{z}$ grid to calculate q and w variance, skewness, kurtosis and temperature and moisture fluxes allows for a better representation of higher order moments, especially at the top of the boundary layer, when compared to the higher order moments derived from the values over the entire domain. By transforming data to a normalized grid, we overcome limitations of the regular grid, particularly during the rapid growth of the CBL. This results in more accurate moments and is more impactful for higher-order (e.g., 3rd) moments. This opens the ability to describe these moments more accurately in a growing CBL, which will lead to improvements in modeling mixing in future climate and weather models.

In forthcoming work, we will discuss methods for handling spatial variability by determining optimum spacing and number of columns to represent a larger domain more accurately. Additionally, work needs to be done to determine optimum analysis





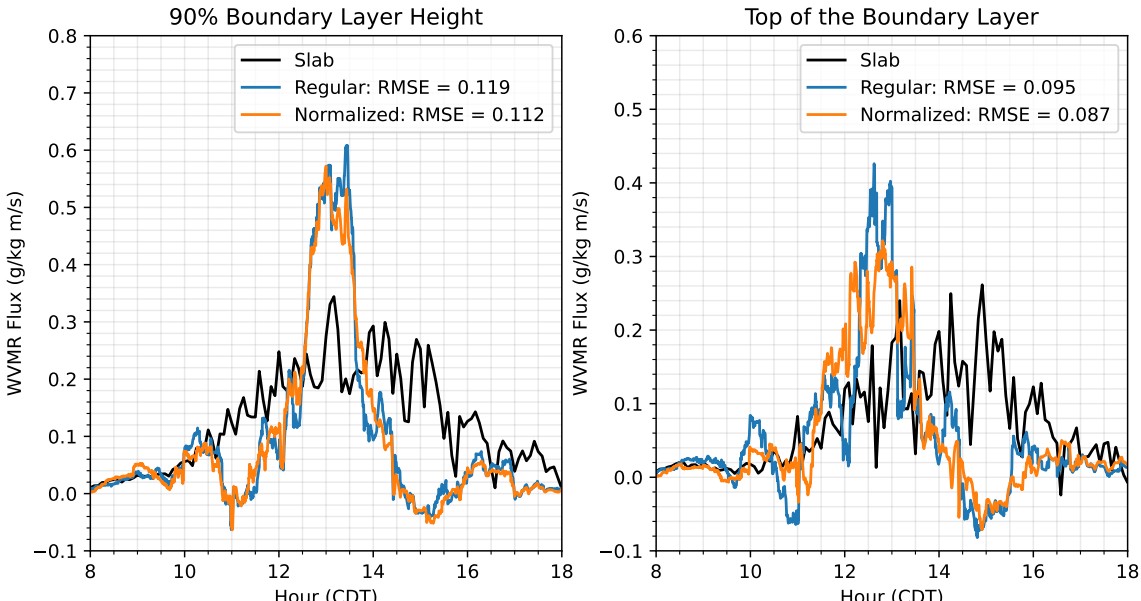

**Figure 10.** Line plot comparison of q flux $(\overline{q'w'})$ [units of $(gkg^{-1})(ms^{-1})^{-1}$] at 90% of the boundary layer (a) and at the top of the boundary layer (b) for the three computation methods: slab (black), regular grid (blue), and normalized $\hat{z}$ grid (orange), along with their respective Root Mean Square Error with respect to the slab value results.

periods and to refine the method for cases where the boundary layer depth is rapidly changing (e.g., during the evening transition).

*Code and data availability.* Code will be uploaded to a GitHub repository before the final review.

*Author contributions.* TH and DT conceived the concept, which was further advanced by all authors. TR performed the LES runs and analysis. TR wrote the manuscript draft, and all authors reviewed and edited the manuscript.

*Competing interests.* The authors declare that they have no conflict of interest.

*Acknowledgements.* This project was supported in part by the U.S. Department of Energy's (DOE) Atmospheric System Research (ASR), and Office of Science Biological and Environmental Research program, under (DE-SC0020114; DE-SC0024048) and the National Oceanic



and Atmospheric Administration's (NOAA) Global System Laboratory (89243019SSC000034) and Atmospheric Radiation Measurement (ARM) program, as well as by the NOAA Atmospheric Science for Renewable Energy Program. Data were obtained from the Atmospheric Radiation Measurement (ARM) User Facility, a U.S. Department of Energy (DOE) Office of Science user facility managed by the Office of Biological and Environmental Research.



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
