# Peer review of "Improving the Estimate of Higher Order Moments from Lidar Observations Near the Top of the Convective Boundary Layer"

_EGUsphere, 2024_

## Referee Comment (RC1)

In the manuscript egusphere-2024-868, titled "Improving the Estimate of Higher Order Moments from Lidar Observations near the Top of the Convective Boundary Layer", the authors test a new approach of computing profiles of higher order moments and fluxes of atmospheric constituents. Higher order moment and flux profiles have been derived by a number of researchers from ground-based lidar observations and they are typically computed on a regular altitude grid in m above ground level. To reduce the sampling errors, higher-order moments and fluxes are normally averaged over a 1-hour time period. When the mixed layer height ($z_i$) is changing rapidly (e.g., during the morning), a 1-hour window near the top of the mixed layer could encompass observations from both, the lower free troposphere and from within the boundary layer, leading to potential biases in the retrieved higher order moments and fluxes. To avoid this, the authors propose to compute the moments and fluxes on an altitude grid normalized by mixed layer height ($z/z_i$), thereby guaranteeing that the data used for moment and flux calculations are completely from within or outside the mixed layer. The authors test the feasibility of this approach by running LES simulations over a 10 km x 10 km domain at the ARM SGP site. The domain-averaged higher order moment and flux profiles at each simulation time step are considered the truth and are compared to 1-hour time averaged moments and fluxes computed on a standard altitude grid and a normalized grid. The authors present convincing evidence that at 0.9 $z_i$ and above the moments and fluxes computed on the normalized grid consistently better match the domain averages than the moments and fluxes computed on the standard grid. The manuscript only contains results for a single day, but the authors state that they ran additional simulations, which produced similar results.

The manuscript is logically well organized and in general written clearly and concisely. The conclusions presented in the manuscript are supported by the data. Tables and figures are all necessary, and the topic of the paper fits well within the scope of AMT. **I recommend publication after minor revisions.**

Comments (suggested changes in bold):

Line 25: "Wulfmeyer et al (2016)" is repeated.

Line 47: "… **restrict** the analysis …"

Figure1, caption: "…  from the slab values (**black**), … of the single column (**orange**)."

Line 56: "… the overbar **indicates** a temporal …"

Line 64: Specify the time step of the LES simulation. It is mentioned in the caption of Fig. 1 ("… instantaneous 10-s values…"), but also spell it out here.

Line 65: "… time series **at** a single location …"

Line 72: Specify the degree of the smoothing polynomial of the Savitzky-Golay filter function. A filter window of 1 hour is mentioned here, but the Fig.1 caption states a 30-min temporal average. Please reconcile.

Line 75: "… cross sections of variance, skewness, **and kurtosis** of water vapor …"

Lines 75-79: State here over which time window the higher order moments and fluxes are averaged. Fig. 2, 4, 6 captions mention "1-h period centered on each 30-min", so I assume the averaging window is 1 hour. Are the moment and flux data produced at the original 10-s resolution by averaging over 1 hour in a gliding fashion? The slab values are available at 10-s resolution. The q variance, skewness, and flux line plots in Figs. 8-10 are plotted at a resolution coarser than 10 s. What is the moment and flux time resolution?

Line 79: What is the $z^\wedge$ grid resolution?

Line 90: "… of the variance, skewness, **and kurtosis** than using …"

Lines 90-91: Discuss briefly the error contribution of interpolating the time series data onto the $z^\wedge$ grid and interpolating the higher order moment and flux profiles back onto the z grid for comparison purposes.

Line 92: **3. Results**

Lines 93-94: … are **the** truth to which …"

Lines 96-98: "… but the normalized $z^\wedge$ grid has less of a gap just before 1500 CDT while the regular grid has a more significant gap there. This tells us that the normalized $z^\wedge$ grid captures the variance better than the regular grid." The differences between the regular and normalized grid q variances appear so small, that this statement is not justified based on the data shown in Fig. 2. Fig. 8 quantifies the slight improvement at 0.9 and 1.0 zi that is gained by using a normalized grid. Reserve the statement about which grid type better captures the true q variance until after Figs. 2, 3, 8, and Table 2, 3 have been discussed.

Line 98: It seems Fig. 3a and 3b are exactly identical. Do they both show 2c – 2a?

Line 99: "… values **except at** 12:30 CDT …"

Lines 111-113: "… that both methods match  the slab values **quite well** in the early morning and in the late afternoon, but not  from 1000-1730 CDT, except for right along the top of the boundary layer.

Line 116: State more clearly that RMSE refers to the RMSE of the difference between the grid and slab values.

Lines 121-122: "… at the flux (Figure 10), **the normalized z^ grid method yields slightly smaller RMSE values at 90% of the boundary layer (Figure 10a) and at the top of the boundary layer (Figure 10b).**

Lines 126-127: "In these tables, **lower values are bolded **"

Line 128: "… or the values were better for an equal number of calculations …" Unclear what the authors mean by that.

Line 131: "… except w **variance**, where the methods **yield the same RMSE.**

Lines 131-132: Finally, at the top of the boundary layer (Table 3), the normalized z^grid method is better in all cases.  Fig. 8b shows that normalized RMSE is lower (0.391) than regular RMSE (0.403). It appears that the former value has not been entered correctly into Table 3.

Lines 132-133: "At every height, the normalized z^ grid method was better for q skewness." This is also true for w skewness, w kurtosis, and both fluxes.

Line 135: **4. Discussion**

Lines 147-148: "…we must remember **that a single column will never be able to properly capture the spatial variability because of sampling errors**. It is clear, especially in the **q variance and q flux time height cross sections** around 1230 CDT…"

Line 150: **5. Conclusions**

Line 153: "… to the higher order moments **and fluxes** derived …"

*Figures and Tables:*

Group all three tables together after Fig. 10.
Q flux unit: (gkg^-1) (ms^-1)
T flux unit: (K) (ms^-1)
Bold z^ RMSE values for w variance and w skewness in Table 1.
I gather that the line plots in Figs. 8-10 are extracted from the time-height cross sections that are all ona regular grid (see Figs 2, 4, 6). Do the line plots represent data at the regular grid height that is closest to 0.9/1.0 zi?

*References:*

Lines 198-199: Lenschow et al. (2000) is cited incorrectly. Substitute with:
"Lenschow, D. H., Wulfmeyer, V., and Senff, C.: **Measuring second- through fourth-order moments in noisy data.,** J. Atmos. Oceanic Technol., 17, 1330–1347, https://doi.org/10.1175/1520-0426(2000)017<1330:MSTFOM>2.0.CO;2., 2000."

---

## Author Comment (AC1)

Thank you for your thoughtful reading of our work and for all of your suggestions and comments. Our responses are below in blue font.

In the manuscript egusphere-2024-868, titled "Improving the Estimate of Higher Order Moments from Lidar Observations near the Top of the Convective Boundary Layer", the authors test a new approach of computing profiles of higher order moments and fluxes of atmospheric constituents. Higher order moment and flux profiles have been derived by a number of researchers from ground-based lidar observations and they are typically computed on a regular altitude grid in m above ground level. To reduce the sampling errors, higher-order moments and fluxes are normally averaged over a 1-hour time period. When the mixed layer height ($z_i$) is changing rapidly (e.g., during the morning), a 1-hour window near the top of the mixed layer could encompass observations from both, the lower free troposphere and from within the boundary layer, leading to potential biases in the retrieved higher order moments and fluxes. To avoid this, the authors propose to compute the moments and fluxes on an altitude grid normalized by mixed layer height ($z/z_i$), thereby guaranteeing that the data used for moment and flux calculations are completely from within or outside the mixed layer. The authors test the feasibility of this approach by running LES simulations over a 10 km x 10 km domain at the ARM SGP site. The domain-averaged higher order moment and flux profiles at each simulation time step are considered the truth and are compared to 1-hour time averaged moments and fluxes computed on a standard altitude grid and a normalized grid. The authors present convincing evidence that at 0.9 $z_i$ and above the moments and fluxes computed on the normalized grid consistently be[er match the domain averages than the moments and fluxes computed on the standard grid. The manuscript only contains results for a single day, but the authors state that they ran additional simulations, which produced similar results. The manuscript is logically well organized and in general wri[en clearly and concisely. The conclusions presented in the manuscript are supported by the data. Tables and figures are all necessary, and the topic of the paper fits well within the scope of AMT. I recommend publica0on a1er minor revisions. Comments (suggested changes in bold):

Line 25: "Wulfmeyer et al (2016)" is repeated. Corrected

Line 47: "… restrict the analysis …"  now, line 49: we have adjusted the phrasing to say "restricting the analysis to shorter time periods"

Figure1, caption: "… from the slab values (black), … of the single column (orange)." Corrected

Line 56: "… the overbar indicates a temporal …" Corrected

Line 64: Specify the time step of the LES simulation. It is mentioned in the caption of Fig. 1 ("… instantaneous 10-s values…"), but also spell it out here. Done

Line 65: "… time series at a single location …" Corrected

Line 72: Specify the degree of the smoothing polynomial of the Savitzky-Golay filter function. A filter window of 1 hour is mentioned here, but the Fig.1 caption states a 30-min temporal average. Please reconcile.

The Savitzky-Golay filter used the 3[rd] order degree, and it was 1-h. The caption to Figure 1 has been corrected.

Line 75: "… cross sections of variance, skewness, and kurtosis of water vapor …" Corrected

Lines 75-79: State here over which time window the higher order moments and fluxes are averaged. Fig. 2, 4, 6 captions mention "1-h period centered on each 30-min", so I assume the averaging window is 1 hour. Are the moment and flux data produced at the original 10-s resolution by averaging over 1 hour in a gliding fashion? The slab values are available at 10-s resolution. The q variance, skewness, and flux line plots in Figs. 8-10 are plo[ed at a resolution coarser than 10 s. What is the moment and flux time resolution? The slab values

are output at 5-minute resolution, which has been corrected in the methods section (line 64). This is the resolution at which the slab is plotted. The single column output is at 10-s, and those data are used in the calculations. All the calculations are done using a running average over 1-hour. The resultant moment and flux temporal resolution is still 10s, and each of those calculations is done using the surrounding hour of temporal data.

Line 79: What is the $z^\wedge$ grid resolution? There are 300 equispaced levels between the surface and zi.

Line 90: "… of the variance, skewness, and kurtosis than using …" Corrected

Lines 90-91: Discuss briefly the error contribution of interpolating the time series data onto the $z^\wedge$ grid and interpolating the higher order moment and flux profiles back onto the z grid for comparison purposes. Since the $z^\wedge$ grid is so much finer in resolution than the z grid, the errors due to interpolation are extremely small. We checked this by simply interpolating the q values to the $z^\wedge$ grid and back and then calculating the RMSE of the result. We found that the RMSE was on the order of $10^\wedge$-4 g/kg.

Line 92: 3. Results Corrected

Lines 93-94: … are the truth to which …" Corrected

Lines 96-98: "… but the normalized $z^\wedge$ grid has less of a gap just before 1500 CDT while the regular grid has a more significant gap there. This tells us that the normalized $z^\wedge$ grid captures the variance be[er than the regular grid." The differences between the regular and normalized grid q variances appear so small, that this statement is not justified based on the data shown in Fig. 2. Fig. 8 quantifies the slight improvement at 0.9 and 1.0 zi that is gained by using a normalized grid. Reserve the statement about which grid type be[er captures the true q variance until aher Figs. 2, 3, 8, and Table 2, 3 have been discussed.

Thank you, this is a good point. This discussion has been moved to line 120.

Line 98: It seems Fig. 3a and 3b are exactly identical. Do they both show 2c – 2a? Yes, they were the same; we apologize for that mistake. This has been fixed.

Line 99: "… values except at 12:30 CDT …" Corrected

Lines 111-113: "… that both methods match with the slab values quite well in the early morning and in the late ahernoon, but not as much from 1000-1730 CDT, except for right along the top of the boundary layer., where it is very close to the slab values Corrected

Line 116: State more clearly that RMSE refers to the RMSE of the difference between the grid and slab values We have updated line 120 to say "The RMSE is calculated based on the difference between each grid method and the slab values."

Lines 121-122: "… at the flux (Figure 10), the normalized z^ grid method yields slightly smaller RMSE values at 90% of the boundary layer (Figure 10a) and at the top of the boundary layer (Figure 10b). Corrected

Lines 126-127: "In these tables, lower values are bolded by the standard error of the two or more are bolded to show the better value." The phrasing here was confusing. We have updated it to say "In this table, the grid method with the lower standard error for a given variable is bolded"

Line 128: "… or the values were better for an equal number of calculaions …" Unclear what the authors mean by that. We mean that, the regular grid was better for 2 cases (e.g., q variance and temperature flux), the normalized grid was better for 2 cases (e.g., q skewness

and water vapor flux), and the rest were identical RMSE values. So, we can't say that, at 0.5 zi, one method was better than the other overall. Additionally, there is no clear advantage to using one method versus another for the calculations. (e.g., if the normalized was better for both skewness calculations). At 0.5zi, the two methods yield equivalent results.

Line 131: "… except w variance, where the methods yield the same RMSE. Corrected

Lines 131-132: Finally, at the top of the boundary layer (Table 3), the normalized z^grid method is be[er in all cases. except q variance, where the methods are the same. Fig. 8b shows that normalized RMSE is lower (0.391) than regular RMSE (0.403). It appears that the former value has not been entered correctly into Table 3. Thank you for catching that mistake. It has been corrected.

Lines 132-133: "At every height, the normalized z^ grid method was be[er for q skewness." This is also true for w skewness, w kurtosis, and both fluxes. While the values from the normalized grid are consistently lower, we wanted to only say that one value was "better" than another when it was lower than the other by the standard error of the two. For this reason, we did not bold each time one value was lower, and the w skewness was not identified as being better for all heights. However, you are absolutely right about the w kurtosis and the fluxes. This has been amended.

Line 135: 4. Discussion

Lines 147-148: "…we must remember that a single column will never be able to properly capture the spa0al variability because of sampling errors. It is clear, especially in the q variance and q flux time height cross seconds around 1230 CDT…" Corrected

Line 150: 5. Conclusions Corrected

Line 153: "... to the higher order moments and fluxes derived ..." Corrected

Figures and Tables:

Group all three tables together after Fig. 10. Done

Q flux unit: (gkg^-1) (ms^-1) Corrected

T flux unit: (K) (ms^-1) Corrected

Bold $z^\wedge$ RMSE values for w variance and w skewness in Table 1. These values are not lower by the standard error of the two methods, so they are not bolded.

I gather that the line plots in Figs. 8-10 are extracted from the time-height cross sections that are all ona regular grid (see Figs 2, 4, 6). Do the line plots represent data at the regular grid height that is closest to 0.9/1.0 $z_i$ Yes

References: Lines 198-199: Lenschow et al. (2000) is cited incorrectly. Substitute with: "Lenschow, D. H., Wulfmeyer, V., and Senff, C.: Measuring second- through fourth-order moments in noisy data., J. Atmos. Oceanic Technol., 17, 1330–1347, h[ps://doi.org/10.1175/1520-0426(2000)017<1330:MSTFOM>2.0.CO;2., 2000. Corrected

---

## Author Comment (AC2)

This work of Rosenberger et al. discusses a new approach to analyze higher-order moments of turbulent fluctuations in the convective boundary layer (CBL). Instead of normalizing the height range against the mean depth of the CBL in the analysis period (typically 30 minutes to 2 hours), the authors suggest a height grid normalized with a temporally smoothed depth of the CBL.

The authors base their analyses on large eddy simulation (LES) data of one day and discuss variance, skewness, and kurtosis of the turbulent fluctuations of water vapor mixing ratio as well as of vertical wind, and of the covariance of these two quantities, the latent heat flux (water vapor flux). Furthermore, they compare these simulated measurements of a vertical pointing lidar (one column measurements) with the data averaged over the whole domain of their model.

The manuscript is well written. I recommend accepting the manuscript after minor revision.

Specific comments:

Page 4, line 70ff: I think it would help if the authors added formulas to explain how they calculated the parameters for the three methods (more than just text). In addition, I suggest adding example plots of the three methods (or at least of the two lidar simulations) for a 1-hour example.

We have added more context about the derivation of the three methods of boundary layer depth calculations and for calculating the level of neutral buoyancy definition of boundary layer depth. Figure 1 shows an example of the three different methods. We have tried to make the connection thereof clearer by adding "All three of these were derived as the level of neutral buoyancy where the $\theta$ used for $\langle z_i(t) \rangle$ was from the slab-averaged LES output, for $z_{c,i}(t)$ it was the instantaneous $\theta$ from an individual column, and $\overline{z_{c,i}(t)}$ is the temporally averaged $z_{c,i}(t)$." to line 75.

Introduction: Please add references to the statements in the first paragraph. Done

Figure 1: Please use the same nomenclature for the different parameters like in the text (line 74). 'Blue' should be 'cyan', 'green' is 'black'. Mark local noon, sunrise, and sunset.

We apologize for the inconsistency. Thank you for your careful reading. The requested timings have been added to the figures in gray dashed lines.

All figures: Please add labels (a), (b), (c).

Done

Figure 2ff: Please define the white areas. I would prefer units at all color bars (instead of explaining the units of the color bars in the figure captions) and labels with the full parameter information (not just "Variance", "Flux" etc.).

This is an excellent suggestion for more clarity. We have added "In each of the following contour figures, data above 1.2 $z_i$ has been masked so that we can focus on the top of the boundary layer and below." in line 96 to explain the white areas, and the colorbar labels have been updated.

Table 1, 2, 3: You refer to RSME values but do not show these. I suggest that you also show errthe RSME data in the tables.

Thank you for the comment. The values listed in the tables are the RMSE values, we have added "Table 1 A-C compares the RMSE values" in line 128 for clarification.

I think it would be better to use the same color scales for all plots of a figure, not different ones, so that it is easier to compare the results (Figs. 2, 6, 7). Same for the y scales in Figs. 8, 9, 10.

Thank you for your suggestion, however we feel that by using the same y scales across different moments, we will lose some of the information we wish to see in comparing the three different calculation methods. Since we are not comparing higher order moment results to one another (i.e., skewness to variance), we think that it is best to keep the scales consistent across the individual moment only (i.e., all the variance ranges are the same for the three different methods) rather than across the different ones.

I would also prefer time scales with ticks at 6, 12, 18 h etc.(not at 5, 10, 15, 20 h). In addition, small ticks at each hour would be helpful.

Thank you, this is much clearer and has been updated.

Instead of "error", I would prefer the term "uncertainty". We changed "sampling error" to "sampling uncertainty."

Instead of "q flux" (Fig. 6) or "WVMR flux" (Fig. 10), I would prefer "latent heat flux".

Thank you for the suggestion, we have also changed the potential temperature flux to sensible heat flux to keep it consistent.

All units should be in normal font (not italic).
Done

---

## Author Comment (AC3)

The authors would like to thank all the reviewers for their careful reading and thoughtful comments on this paper. Our responses are in blue below.

Public reviewer comments

This manuscript proposes a methodology to improve the derivation of boundary-layer statistics from lidar observations. To do so, they propose to use the information derived from the observations but normalized by the boundary layer height before time-averaging. Although the subject is interesting, some modifications are needed before the manuscript is in the form for the definitive publication.

Introduction:
The introduction is well written, however a statement of the main objectives of the paper and the introduction of the outline are lacking. Thank you for the observation. We have added "This work presents a new approach (outlined in Section 2) to analyze lidar profile observations over time when the height of the CBL is changing over that time. The approach is simple: change the vertical coordinate from height to normalized height before computing the statistics over temporal windows. This paper demonstrates this approach using output from a large eddy simulation model (Section 3), wherein we can use a single column to approximate the lidar observations and spatial statistics to serve as truth." at line 55.

Approach:
I suggest to repeat the same analysis, as shown in the paper, but for a certain amount of different locations sampled in the LES (instead of relying on outputs at only one given location) and then compute statistically the mean rmse of the difference between the computation over the regular grid and the one over the normalized grid to statistically demonstrate the improvement. The fact that the results rely on only one high-frequency time-serie is not completely convincing. Thank you for your suggestion. We conducted this analysis for five separate dates and calculated the average RMSE across those five individual dates. The results of which are included in a table at the end of this discussion. We see that the normalized method has the smaller RMSE than the regular method on average across the five dates.

Results:
I propose to reduce slightly the number of figures. This can easily be done by combining Figures 2 and 3: you could just show the time-height variance for the slab and then only show the difference between the regular and slab and between the normalized and slab. Similarly for figures 4 and 5 and Figures 6 and 7. Thank you for your suggestion. We have removed the difference plots altogether, as we think our message comes through with the comparison plots alone.

Minor comments:
l 69: please change 'derived three different ways' to 'derived through three different ways' Changed "ways" to "methods."
l 92: 'Results' should be a section and not a subsection Corrected

| | Average over 5 days 75% zi | |
|---|---|---|
| Moment | Regular | Normalized |
| Variance q' | **0.087** | 0.089 |
| Skewness q' | 0.661 | **0.515** |
| Kurtosis q' | **1.297** | 1.329 |
| Variance w' | 0.168 | 0.168 |
| Skewness w' | 0.539 | **0.523** |
| Kurtosis w' | 2.024 | **1.950** |
| Wvmr flux | 0.0534 | **0.0528** |
| Thl flux | 0.0176 | **0.0165** |
| | 90% zi | |
| Variance q' | 0.209 | **0.199** |
| Skewness q' | 0.766 | **0.675** |
| Kurtosis q' | 3.703 | **3.537** |
| Variance w' | 0.122 | **0.117** |
| Skewness w' | 0.563 | **0.537** |
| Kurtosis w' | 2.469 | **2.375** |
| Wvmr flux | 0.0725 | **0.0716** |
| Thl flux | 0.0295 | **0.0286** |
| | zi | |
| Variance q' | 0.358 | **0.357** |
| Skewness q' | 1.164 | **1.053** |
| Kurtosis q' | 13.431 | **13.110** |
| Variance w' | 0.108 | **0.102** |
| Skewness w' | 0.636 | 0.640 |
| Kurtosis w' | 2.541 | **2.501** |
| Wvmr flux | 0.075 | **0.074** |
| Thl flux | 0.037 | 0.037 |